

# Purification and characterization of detergent stable alkaline lipase from *Bacillus safensis* TKW3 isolated from Tso Kar brackish water lake

Tishu Devi[1], Srinivas Sistla[2], Rabiya T. Khan[1], Swadha Kailoo[1], Mansavi Bhardwaj[1] and Shafaq Rasool[1]

[1] School of Biotechnology, Shri Mata Vaishno Devi University, Katra, J&K, India
[2] Microbiology and Immunology Department, State University of New York, Stony Brook, New York, United States of America

## ABSTRACT

Extensive and escalating research has been directed towards halozymes derived from halophiles thriving in extreme hypersaline environments, owing to their myriad industrial applications. These extremophiles have evolved various physiological and metabolic adaptations to endure such extremes, enhancing their industrial potential. Being a potential source of lipases, halophiles of extreme niches have emerged as a emerging research area. This interest has been fueled by the recognition that extreme environments serve as rich reservoirs of diverse cold-active alkaliphilic enzymes.

**Methods:** *Bacillus safensis* TKW3, isolated from brackish Lake Tso Kar of the Ladakh region, India, produced cold-adapted haloalkaliphilic lipase halozyme. The current study focused on the purification and biochemical characterisation of lipase derived from halophilic bacteria.

**Results:** The lipase enzyme, purified to homogeneity, exhibited a molecular mass of 28 kDa as confirmed by SDS-PAGE analysis. The purification process yielded a purification fold of 12.01 and a final recovery rate of 29.9%. It demonstrated optimal activity at 30 °C and pH 9. The enzyme was evaluated and demonstrated to exhibit stability over a broad temperature range spanning from 5 °C to 55 °C, as well as a wide pH range of 7.0 to 9.0. Due to its stability across a diverse spectrum of pH values, surfactants, metal ions, and inhibitors, the enzyme appeared to hold significant promise for application within the leather and detergent sectors. Upon undergoing detergent compatibility tests spanning diverse temperature ranges, the lipase showcased compatibility with various commercial detergents, thereby presenting itself as an attractive candidate for inclusion in detergent formulations within the industry.

**Conclusions:** The lipase from *B. safensis* TKW3 exhibits promising attributes, including alkali stability, halophilicity, and a wide spectrum of substrate specificity, rendering it an attractive option for incorporation into detergent formulations within the detergent industry. As far as we are aware, this is the first report on the purification and characterization of lipase enzyme from bacterial halophiles in a Tso Kar brackish lake.

Corresponding author
Shafaq Rasool,
shafaq.rasool@smvdu.ac.in

## INTRODUCTION

The prevailing reality of industrial processes often involves harsh conditions that cannot consistently align with the optimal parameters required for enzyme activity (*Kamekura, 1986*). Enzymes derived from extremophiles possess the remarkable ability to withstand harsh environmental conditions. Their unique characteristics make them invaluable resources for pioneering advancements in biotechnological methodologies and applications across various industries. Halophilic bacteria have attracted the interests of researchers due to their adaptability to a wide range of salinities. The extremities of halophiles make them interesting models for fundamental research and exploration of biotechnological potential (*Margesin & Schinner, 2001*; *Rohban, Amoozegar & Ventosa, 2009*). A vast reservoir of novel biomolecules, metabolites, and biomaterials is thought to exist in halophiles. In industrial biotechnological applications, halophiles have the potential to play a wide range of, roles such as the synthesis of exopolysaccharides, biosurfactants, osmoregulants, and, biopolymers during oil recovery; detergent industry enzymes; and bioremediation of contaminated hypersaline brines. The halozymes produced by these halophiles are resilient even in extreme environmental conditions, thus making them apt for various industrial process applications. It has been reported that extreme halophilic bacteria are prospectively being used to treat wastewater (*Alva & Peyton, 2003*; *Peyton, Wilson & Yonge, 2002*). Halozymes, including amylases, proteases, lipases, and others, have emerged as indispensable tools for a myriad of applications across diverse industries (*Mellado et al., 2004*). From food processing to pharmaceuticals, these enzymes play pivotal roles in catalysing biochemical reactions with remarkable efficiency and specificity. Notably, the laundry detergent industry has harnessed the power of halozymes derived from microbial sources, integrating them into detergent formulations to enhance stain removal and fabric care (*Coker, 2016*). Among the pantheon of halozymes, one enzyme stands out for its versatility and significance in industrial processes: lipase. Lipases, with their ability to catalyse the hydrolysis of lipids, find extensive use in various sectors, including biodiesel production, food processing, and pharmaceutical formulation. Lipases sourced from extreme environments have gathered considerable attention as a promising avenue for bioresource technology research (*Sanchez & Demain, 2017*), given the tendency of extreme niches to host a diverse array of cold-active alkaliphilic enzymes. Since halophilic bacteria thrive in high salinity environments (cold, hot, high salinity level, organic solvents), halophilic lipases are typically well suited to extremely hard conditions (*Barone et al., 2014*; *Moreno et al., 2016*). These characteristics are necessary for certain industrial processes, including the manufacturing of biopolymers, biodiesel, bioremediation, and waste treatment. Halophilic lipases can enhance the efficiency and environmental friendliness of the industrial processes (*Bornscheuer, 2002*). These robust biocatalysts can be applied in various industrial and biotechnological processes to enable substrate conversions under extreme harsh process conditions. The Tso Kar, is a heavily saline lake with large heaps of salt

around it that provides an extreme environment for the survival of microbes and can serve as a source for the discovery of extremozymes. Tso Kar is a fluctuating salt lake situated at 33180 N, 78000 E, at about 4,527 m above mean sea level, occupying an area of about 200 km$^2$ in the Rupshu Plateau, about 125 km southeast of Ladakh (*Kramer, Kotlia & Wünnemann, 2014*). It is also known as "White Lake" due to its brackish water and salt build-up. Winter temperatures range from −10 °C to −40 °C due to the high altitude. Low rainfall causes summer temperatures to reach 30 °C (*Philip & Mazari, 2000*).

In the present study, a moderately halophilic bacterial strain, *Bacillus safensis* TKW3, was isolated from the water sample of the Tso Kar brackish water lake. The strain was screened for the presence of lipase enzyme, and it showed good lipolytic activity. The lipase enzyme was purified to homogeneity following a three-step purification protocol. The purified enzyme was biochemically characterized to determine the potential of this lipase enzyme. The compatibility of the enzyme to work with detergents was also determined to infer the suitability of lipase for detergent formulation in the lipase industry.

## MATERIALS AND METHODS

### Bacterial strain and lipase production

TKW3, a moderately halophilic bacterial strain, isolated in the previous study from the Tso Kar Brackish water lake by enrichment method (SupplementaryDataStaffSS.docx), was used for the production of the lipase enzyme. The molecular identification of the strain TKW3 was done with the help of 16S rDNA PCR amplification. The genomic DNA isolation was carried out by the method reported by *Wilson (1989)*. For PCR amplification of 16S rDNA, universal primer sets 27F (5′-AGAGTTTGATCMTGGCTCAG-3′) and 1492R (5′-TAC-GGYTACCTTGTTACGACTT-3′) (*Lane, 1991*) were used. Amplification was done in an automated thermocycler (Eppendorf) with amplification steps as: Preheating at 94 °C for 3 min., followed by 29 cycles of denaturation at 94 °C for 30 s, an annealing step at 55 °C for 1 min, and an extension step at 72 °C for 1 min, followed by final extension at 72 °C for 10 min. The PCR product was purified using a Qiagen PCR Purification kit as mentioned in the manufacturer guidelines, and the purified products were sequenced in both directions. To identify the closest neighbor the generated nucleotide sequences were subjected to a BLAST search in the NCBI GenBank database (http://www.ncbi.nlm.nih.gov). A phylogenetic analysis was carried out by obtaining related sequences from the NCBI, and an evolutionary tree was created using the neighbor-joining analysis method and MEGA software, with 1,000 bootstrap values (*Hall, 2013*). The 16S rDNA sequence of TKW3 strain was submitted to NCBI.

The strain TKW3 was screened for the localization of enzyme activity. Both extracellular and intracellular enzyme fractions were analyzed for enzyme activity.

To determine the optimal conditions for lipase production, one variable at a time maintaining all the factors unaltered was executed to study the influence of various nutrient and physical parameters on the growth and lipase production of the strain TKW3 (Table 1). The basal medium was LB broth with a pH of 7.0 supplemented with 12% of sodium chloride and no additives was used as the basal medium. Growth was determined

**Table 1 Nutrient and physical parameters and levels of one-factor experiment.**

| Factors | Types | Levels | Factors | Types | Levels |
|---|---|---|---|---|---|
| **Carbon sources** | Maltose | 1% | **Nitrogen sources** | Glycine | 1% |
| | Sucrose | | | NaNO$_3$ | |
| | Cellulose | | | Tryptone | |
| | Starch | | | (NH4)2SO4 | |
| **Inducers** | | | **Metal ions** | LiCl2 | 1 mM |
| | Tributyrin | 1% | | MnCl2 | |
| | Mustard oil | | | CaCl2 | |
| | Olive oil | | **Surfactant** | Sarcosyl | 0.5% |
| | Coconut oil | | | SDS | |
| | Ghee | | | Triton X-100 | |
| | | | | Tween-20 | |
| | | | | Tween-80 | |

| Factors | Levels |
|---|---|
| **NaCl** | 1, 2, 3, 4, 5, 6, 7 M |
| **Temperature** | 5 °C, 15 °C, 20 °C, 25 °C, 30 °C, 37 °C, 45 °C |
| **pH** | 2, 4, 7, 8, 9, 10, 12 |

spectrophotometrically at 600 nm, and lipase activity was measured at 405 nm by the hydrolysis of substrate pNP-octanoate.

## Preparation of cell free extract

The TKW3 strain was inoculated in the optimized broth media for enzyme production. The inoculated broth was incubated at 30 °C in an incubator shaker at 181 rpm. After 60 h., the grown culture was centrifuged at a speed of $10,000 \times g$ for 30 min to collect the pellet for obtaining the intracellular fraction containing the lipase enzyme. The cell pellet was resuspended in 50 mM phosphate buffer (pH 7), followed by disruption of cells at 4 °C using a Q-Sonica sonicator. The cell pellet was sonicated for five cycles with a 1-minute pulse at 15 kHz followed by a 30-s pause. The disrupted cell suspension was then centrifuged at $10,000 \times g$ for about 30 min at 4 °C to collect the cell free extract (CFE) as an intracellular enzyme.

## Purification of lipase enzyme

To purify the lipase enzyme from the cell free fraction, a three-step purification protocol was followed: ammonium sulfate fractionation followed by dialysis and hydrophobic interaction chromatography (HIC). The intracellular fraction was partially precipitated using 30–75% saturation of ammonium sulfate. After this step, the active fractions were pooled and dialyzed against 50 mM Tris-HCl (pH 8.8) to remove salts. The protein solution was placed in a dialysis membrane, sealed tightly at both ends, and immersed in de-ionized water. The solution was agitated for 16 to 24 h at 37 °C and repeated thrice to

reduce the salt concentration to minimum. The dialysate was then ultra-filtered using 5 kDa molecular weight filters.

The retentate were then loaded onto a Hiload 16/10 Phenyl-Sepharose High-Performance Column (GE Healthcare Lifesciences, Chicago, IL, USA) pre-equilibrated with 50 mM Tris-HCl of pH 8.8 containing 1.0 M ammonium sulfate. The flow rate through the column was set at 1.5 ml/min. The adsorbed proteins were eluted using a decreasing linear gradient of ammonium sulfate (1 to 0 mol/L) in 50 mM Tris-HCl (pH 8.8), and the active fractions were mixed. The combined fractions were dialyzed against 20 mmol/L Tris-HCl (pH 8.0), and concentrated using 5 kDa ultrafiltration. The resulting fractions were evaluated for lipase enzyme activity using the spectrophotometric approach at a wavelength of 405 nm, employing p-nitrophenyl octanoate as the substrate, and the protein concentration was determined using the Bradford method (*Bradford, 1976*).

## Determination of molecular weight

The purified protein was separated using a 12% SDS-PAGE gel in accordance with the method described by *Laemmli (1970)*, and the molecular weight of the enzyme was determined using a marker with a standard protein molecular weight ranging from 20.1 to 205 kDa (GeNei).

## Determination of lipase activity by zymogram

The lipase activity of the purified protein was assessed using 10% native page electrophoresis under non-denaturing conditions, following the methodology described by *Febriani, Hertadi & Madayanti (2013)*. After electrophoresis, the gel was incubated in a solution containing 50 mM Tris buffer (pH 8.0) and 1% triton X-100 for 4 h. After incubation, the gel was rinsed in distilled water, and re-immersed in a solution containing 50 mM Tris buffer (pH 8.0), 30 mg/ml naphthyl acetate and 20 mg/ml fast blue RR salt. The gel was kept in the dark for thirty minutes at a temperature of 37 °C and the lipase activity was observed as distinct red bands on the gel.

## Biochemical characterization of the purified lipase

The purified lipase enzyme was characterized with respect to optimum temperature, temperature stability, pH optima, pH stability, and the effects of various parameters such as salts, additives, detergents, and urea. The assays for all the experiments were carried out using 1 mM pNP octanoate as the substrate, with the release of pNP measured at 405 nm under standard conditions (Fig. S1). One unit of lipase activity is defined as the amount of enzyme that can produce 1 $\mu$mol of pNP per minute. All experiments were done in triplicates.

## Effect of temperature on enzyme activity and stability

The optimal temperature for enzyme activity of the purified lipase was estimated spectrophotometrically at 405 nm with pNP octanoate as substrate across various temperatures 5–80 °C. The stability of the enzyme was assessed by incubating the enzyme with Tris buffer, pH 8, for 30 min at different temperatures (5–80 °C), and the remaining enzyme activity was quantified.
### Effect of pH on enzyme activity and stability

The optimum pH of the purified lipase enzyme was determined by estimating pNP release at different pH values (4.0–12.0). Different buffer systems were utilized for each pH range such as: 0.1 M sodium citrate (pH 4.0–5.0), 0.1 M phosphate buffer (pH 6.0–8), 0.1 M Tris-HCl (pH 7.0–10.0), and 0.1 M glycine-NaOH (pH 11.0–12.0). The pH stability of the purified enzyme was evaluated by pre-incubating the enzyme for 30 min in corresponding buffers of different pH values (4.0–12.0). The residual activity was assessed using established enzyme assay protocol (*Massadeh et al., 2012*).

### Effect of salt on enzyme activity and stability

The optimum salt concentration of the purified lipase enzyme was investigated by performing enzymatic hydrolysis reactions at varying salt concentrations ranging from 0.5 to 6 M NaCl. The stability of the enzyme was examined by incubating the purified enzyme in assay buffer for 30 min at 30 °C using pNP octanoate as substrate containing different salt concentrations ranging from 0.5–6 M.

### Effect of surfactants, inhibitors, and metal ions on the activity of the enzyme

The impact of surfactants including SDS, Sarcosyl, Tween-80, Tween-20, and Triton X-100, on the purified lipase enzyme was assessed by incorporating them into the enzyme solution at a concentration of 0.5%. The measurement of enzyme activity was carried out after 30 min of incubation. The effect of inhibitors was evaluated spectrophotometrically after incubating the enzyme with 2 and 5 mM of EDTA, DTT, β-mercaptoethanol, and guanidine hydrochloride for 30 min.

Furthermore, the effect of metal ions on lipase activity was assessed by pre-incubating the purified enzyme with $Ca^{2+}$, $Mg^{2+}$, $Cu^{2+}$, $Li^+$, and $K^+$ at a concentration of 1 mM for 30 min at 30 °C. The residual activity was determined under standard assay conditions. The enzyme's relative activity was calculated by comparing it to an enzyme that was incubated under similar conditions, but without the presence of surfactants, inhibitors, and metal ions (as control, relative activity = 100%).

### Substrate specificity of purified lipase

The substrate specificity of the lipase enzyme was determined spectrophotometrically at 405 nm by using 1 mM pNP esters of different acyl chain lengths, such as p-nitrophenyl esters of acetate ($C_2$), butyrate ($C_4$), caproate ($C_6$), octanoate ($C_8$), decanoate ($C_{10}$), and myristate ($C_{14}$), respectively.

### Detergent compatibility of lipase with commercial detergents and oils

Different commercial detergents were added directly to the enzyme assay mixture to assess the detergent compatibility of the enzyme for use as a detergent additive (*Chellappan, 2006*). Commercial detergents like Tide, Surf excel, and Aerial were used at 1% (v/v) concentrations in order to test the detergent compatibility. The enzymes that were already present in the detergents were first rendered inactive by the use of heat, which was done by

boiling detergents for about 10 min. Detergent solutions were prepared in distilled water, and pNP octanoate was used to quantify the hydrolytic activity. Enzyme without detergents under similar conditions was used as control, and lipase activity in the presence of detergents was used as the test. The assay was carried out for 10 min at various temperatures, specifically 5 °C, 10 °C, 20 °C, 30 °C, 37 °C, and 60 °C.

The lipase from bacterial strain TKW3 was also tested for vegetable oil stain removal on 5 × 5 cm cotton fabric under the following four conditions: The conditions were: a) 100 ml of distilled water (control) b) 100 ml of distilled water and 1 ml of detergent (5 mg/ml) c) 100 ml of distilled water, 1 ml of detergent (5 mg/ml), and 0.05 ml of lipase enzyme d) 100 ml of distilled water, and 0.05 ml of lipase enzyme. The stained cloth was dipped for 30 min. in each condition. After dipping, the clothes were washed with clean distilled water and observed for the intensity of the stain. This helped in evaluating the lipase's efficiency as a detergent additive.

# RESULTS

## Bacterial strain and lipase activity

The generated sequence was submitted to NCBI GenBank under the accession number OP735582.1. The strain TKW3 was identified as *B. safensis*. The phylogenetic tree was also constructed using the MEGA 4.1 program. Homologues rDNA sequences, with a query coverage of either 100% or 99%, were selected from the Blast Local Alignment Search Tool (BLAST), NCBI for the construction of phylogenetic tree (Fig. S2).

On the basis of zones of hydrolysis, it was determined that TKW3 strain has intracellular lipase activity (Fig. S3).

It was determined that optimal conditions for the TKW3 strain to produce the maximum amount of lipase were LB broth with a pH of 9.0, 1% (w/v) sucrose, 1% (w/v) ammonium sulphate, 3 M sodium chloride, 1% (v/v) ghee, 0.5% (v/v) Tween-80, and 1 mM $LiCl_2$. These conditions were maintained for approximately 60 h at 181 rpm at a temperature of 30 °C.

## Production of lipase

The moderately halophilic strain *B. safensis* TKW3 showed intracellular lipase activity. The TKW3 strain exhibited a significant increase in lipase production, reaching around 199.4 IU/ml compared to the initial level of 39.4 IU/ml. This augmentation was achieved by supplementing the growth medium with diverse components, including 1% sucrose, 1% ammonium sulfate, 3 M NaCl, 1% ghee, 0.5% Tween-80, and 1 mM $LiCl_2$ at 181 rpm and 30 °C for approximately 60 h.

## Purification of *B. safensis* TKW3 lipase

A three-step purification protocol was followed, and the lipase enzyme has been successfully isolated and purified from *B. safensis* TKW3. The enzyme was successfully purified to 29.9% achieving a purification fold of 12.01 (Table 2).

**Table 2 Purification summary of lipase from *Bacillus safensis* TKW3.**

| 2 enzyme | Total protein (mg) | Total activity (IU) | Specific activity (IU/mg) | Purification fold | Yield (%) |
|---|---|---|---|---|---|
| Original | 5,186.2 | 206,444 | 39.8 | 1 | 100 |
| 75% precipitation | 165.05 | 74,341.5 | 450.4 | 11.31 | 36 |
| Dialysis | 148.89 | 68,685 | 461.31 | 11.6 | 33 |
| HIC HiLoad 16/10 phenyl-sepharose column | 129.12 | 61,734.88 | 478.12 | 12.01 | 29.9 |

## Determination of molecular weight and zymogram of purified lipase

The purified protein was subjected to electrophoresis using 12% SDS-PAGE to ascertain the molecular weight, demonstrating a singular band around 28 kDa (Fig. 1A). The zymogram analysis further confirmed a single band displaying lipase activity (Fig. 1B).

## Biochemical characterization of the enzyme *B. safensis* TKW3

### Effect of temperature

The optimal temperature for the purified lipase enzyme derived from *B. safensis* TKW3 was found to be 30 °C, while retaining 44.7% activity at 5 °C. The enzyme exhibited remarkable stability over a wide range of temperature ranging from 5 °C to 55 °C; however, a substantial drop in stability of the enzyme was observed after 55 °C (Fig. 2).

### Effect of pH

The purified lipase exhibited its peak activity at pH 9.0, while the activity decreased significantly to about 32% at pH 11.0 and 15% at pH 12.0. The purified lipase enzyme from *B. safensis* TKW3 was found relatively stable in the pH range of 7.0–9.0, where approximately 90% of lipase activity was retained. However, 52% of the activity was retained at pH 10.0. Lipase activity was 1.55-fold higher at pH 9.0 as compared to pH 7.0, with no detectable activity below pH 6, confirming the enzyme's alkaline nature (Fig. 3).

### Effect of salt

In the presence of the salts, the purified lipase enzyme from *B. safensis* TKW3 showed an increase in activity with increasing concentrations from 1 to 3 M. However, enzyme activity started to decline upon further increase in NaCl concentration. A total of 3 M NaCl was identified as the optimal salt concentration for activity of the purified enzyme. Additionally, the enzyme exhibited stability across broad spectrum of salt concentrations, ranging from 0.5 to 6.0 M (Fig. 4).

### Effect of surfactants on enzyme activity

The addition of surfactants resulted in a significant increase in lipase activity, with SDS being the most effective among all the detergents used, followed by Tween-80. When non-iconic detergents such as Triton X-100 and Tween-80 were used, the enzyme activity was enhanced by 47% and 129% respectively. However, a reduction in relative activity of about 26.5% was observed when Tween-20 was used. Interestingly, the presence of strong anionic detergents like SDS and SLS led to a nearly two-fold increase in enzyme activity (Fig. 5).
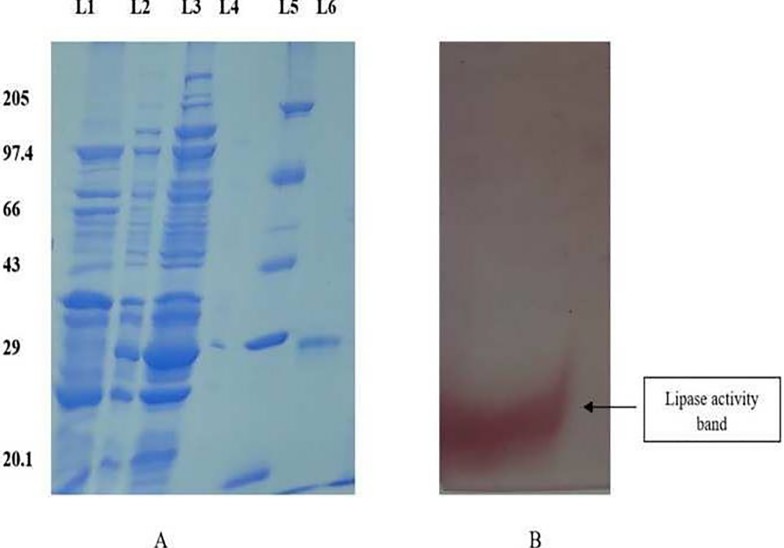

**Figure 1 SDS PAGE profile for determination of molecular weight and native page for determination of lipase activity.** (A) 12% SDS PAGE showing different fractions of purified protein. Lane 1: Dialyzed, Lane 2: 75% ammonium sulphate precipitate, Lane 3: crude, Lane 4: Empty, Lane 5: Molecular weight Marker, Lane 6: Purified fraction. (B) The zymogram analysis of the purified enzyme exhibited distinct red bands on the gel, indicating enzyme activity.

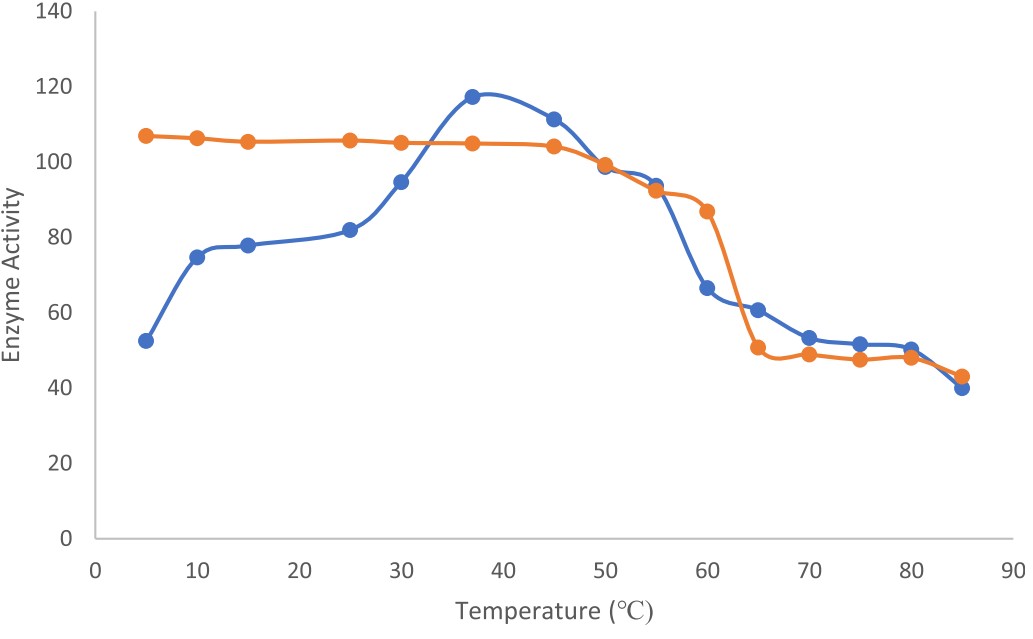

**Figure 2 Effect of temperature on the lipase activity of *B. safensis* TKW3.** All experiments were performed in triplicates and standard deviation was calculated The optimal temperature of the purified lipase enzyme from *B. safensis* TKW3 was determined to be 30 °C. The enzyme exhibited remarkable stability over a wide temperature range, from 5 °C to 55 °C. However, a significant drop in stability of the purified lipase was observed beyond 55 °C.

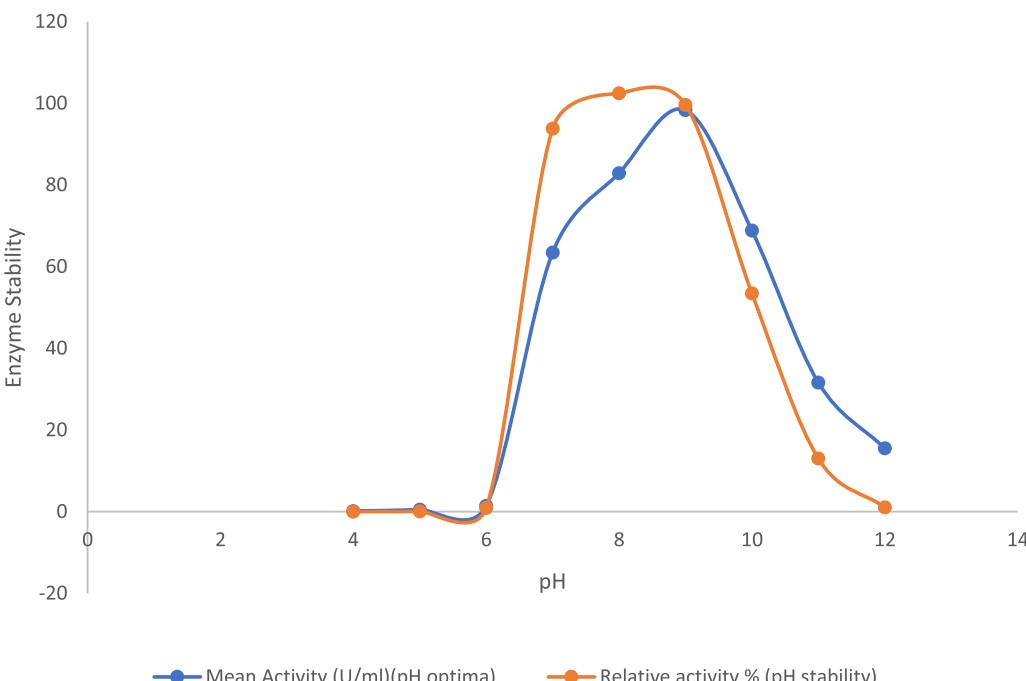

**Figure 3 Effect of pH on the lipase activity of *B. safensis* TKW3.** All experiments were performed in triplicates and standard deviation was calculated. The purified lipase exhibited maximum activity at pH 9.0, with activity decreasing to approximately 32% at pH 11.0 and 15% at pH 12.0. pH stability analysis revealed that the purified lipase from *B. safensis* TKW3 retained about 90% activity within the pH range of 7.0 to 9.0. However, activity decreased to 52% at pH 10.0.

### Effects of inhibitors and metal ions

The effect of inhibitors on the lipase enzyme was determined by the addition of EDTA, ß-ME, DTT, and guanidine hydrochloride to an enzyme substrate mixture at a final concentration of 2 and 5 mM respectively. ß-ME completely reduced the enzyme activity to 6.45% and 1.84% at concentrations of 2 and 5 mM, respectively. DTT was observed to decrease the enzyme activity by 90.3% and 93.5%, respectively. The presence of EDTA increased lipase activity by 55% and 76% at concentrations of 2 and 5 mM, respectively (Fig. 6A).

The effect of various monovalent and divalent cations on lipase activity was assessed. The presence of metal ions showed no effect on the lipase activity of *B. safensis* TKW3 except for $Cu^{2+}$. The presence of the $Cu^{2+}$ ion led to a reduction in the activity of the purified protein, resulting in a decrease of 74% (Fig. 6B).

### Substrate specificity

The various pNPEs with acyl chains of varying lengths were analyzed to investigate substrate selectivity and to identify the optimal substrate for the purified enzyme derived from *B. safensis* TKW3 (acetate, C2; butyrate, C4; hexanoate, C6; caprylate, C8; caproate, C10; dodecanoate C12; myristate, C14). The minimal activity of the short-chain fatty acids resulted in 18% relative initial activity. Nonetheless, the initial hydrolysis of C4 to C6 fatty acid esters was about 87% (Fig. 7).

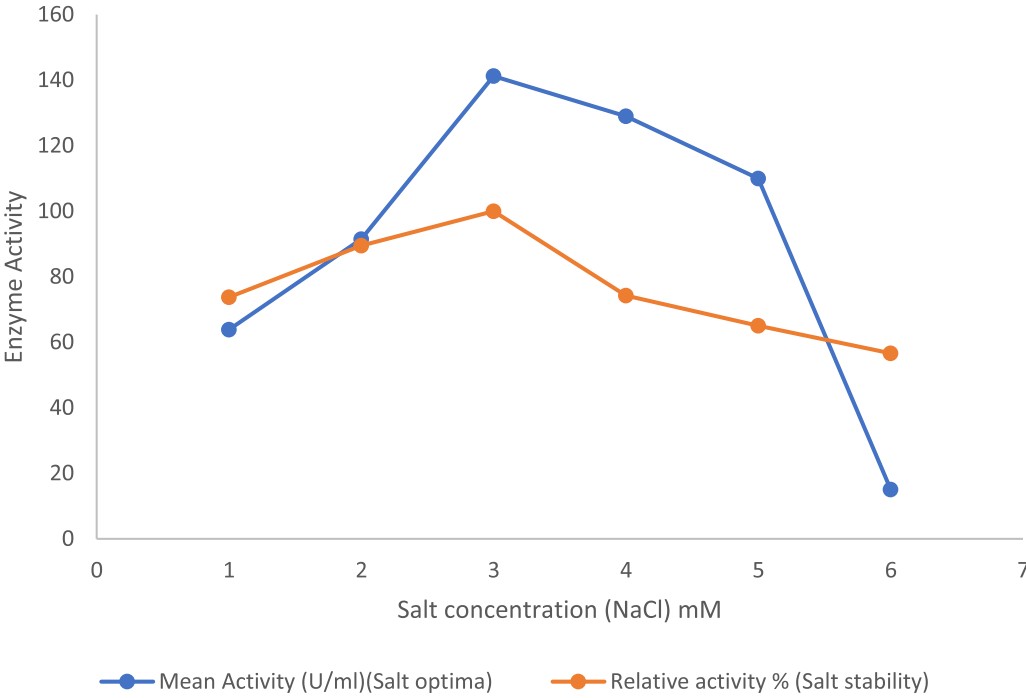

**Figure 4 Effect of NaCl on the lipase activity of *B. safensis* TKW3.** All experiments were performed in triplicates and standard deviation was calculated. The purified lipase enzyme from *B. safensis* TKW3 demonstrated increased activity with rising NaCl concentrations, peaking at 3 M. However, further increases in salt concentration led to a decline in activity. Optimal activity of the purified enzyme was observed at 3 M NaCl concentration. Additionally, the enzyme exhibited stability over a wide range of salt concentrations, ranging from 0.5 to 6.0 M.

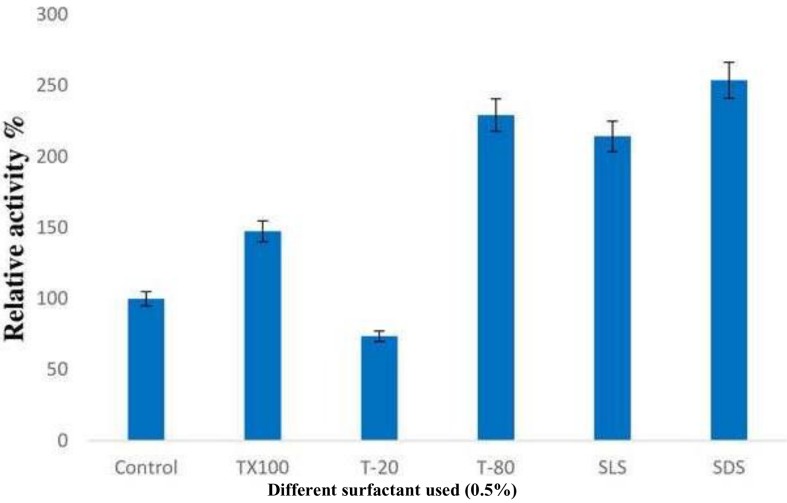

**Figure 5 Effect of surfactants on the enzyme activity of *B. safensis* TKW3.** All experiments were performed in triplicates and standard deviation was calculated The addition of surfactants significantly increased lipase activity. SDS was the most effective surfactant, enhancing enzyme activity the most, followed by Tween-80. The activity increased by 47% and 129% with Triton X-100 and Tween-80, respectively. However, a decrease in activity by approximately 26.5% was observed with Tween-20. Strong anionic detergents like SDS and SLS doubled the enzyme activity compared to the control.

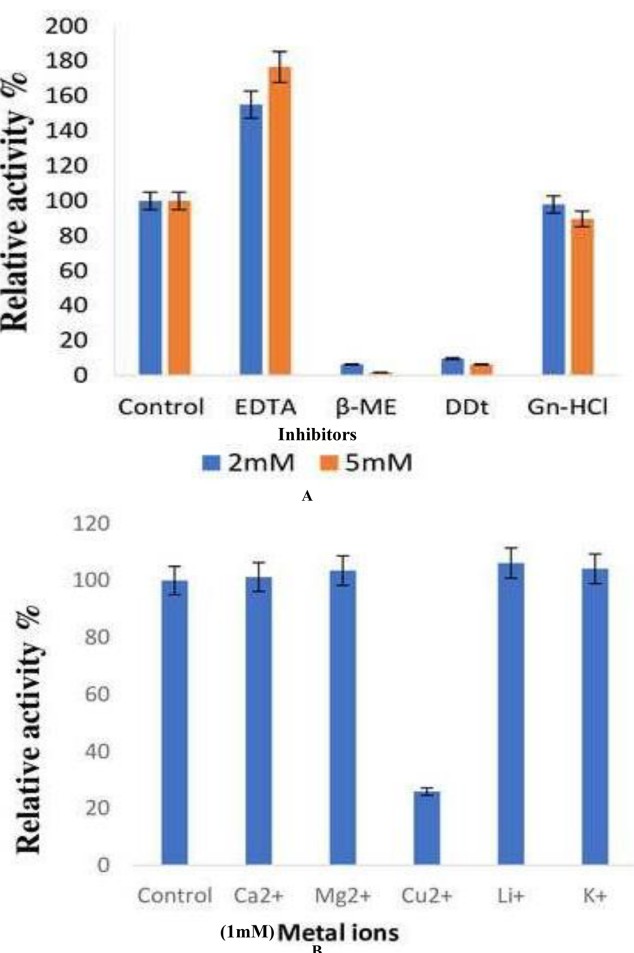

**Figure 6 Effect of inhibitors and metal ions on the enzyme activity of *B. safensis* TKW3.** (A) Effect of inhibitors; (B) effect of metal ions. All experiments were performed in triplicates and standard deviation was calculated Effect of inhibitors on lipase activity. Lipase activity was assayed after incubation with various inhibitors at concentrations of 2 and 5 mM. ß-mercaptoethanol drastically reduced activity, while DTT showed a significant decrease. EDTA exhibited a notable increase in activity at both concentrations. Panel (B) shows the impact of metal ions on lipase activity. Various monovalent and divalent cations were tested for their effect on lipase activity. $Cu^{2+}$ showed a substantial reduction in activity compared to other metal ions, resulting in a 74% decrease in enzyme activity.

## Stability of an enzyme with commercial detergents

Lipolytic enzymes, for use in the demanding detergent industry, must be stable and compatible with all of the common ingredients in detergent formulation. Any lipase that can work with detergent must be stable over a wide range of temperatures. The compatibility of the lipase with the commercial detergent at a wide range of temperatures was studied in order to establish whether or not *B. safensis* TKW3 lipase is a good candidate for incorporation into detergent formulations. The lipase enzyme was pre-incubated for about 30 min at various temperatures (5 °C, 10 °C, 20 °C, 30 °C, 37 °C, and 45 °C) in the presence of various commercial laundry detergents to test the compatibility of the enzyme with different detergents (Figs. 8A and 8B). The lipase activity

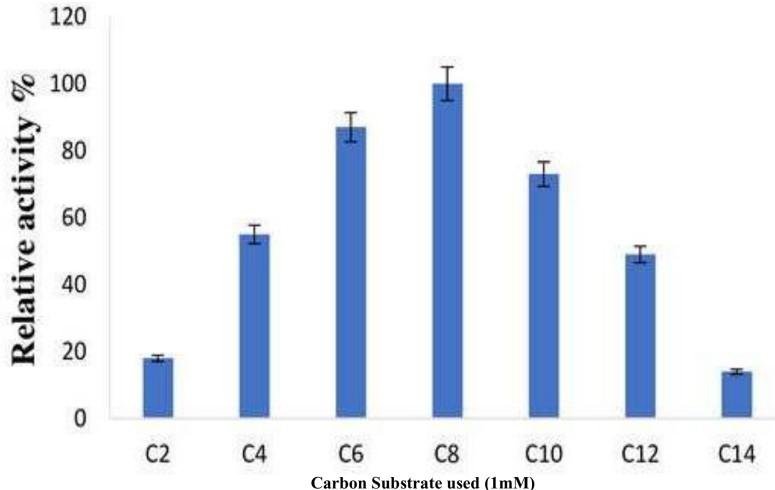

**Figure 7 Substrate specificity of *B. safensis* TKW3.** All experiments were performed in triplicates and standard deviation was calculated Various p-NPEs with acyl chains of different lengths (C2 to C14) were tested to assess substrate preference. Short-chain fatty acids exhibited minimal activity, resulting in 18% relative initial activity. However, the initial hydrolysis of C4 to C6 fatty acid esters reached approximately 87%.

of the enzyme without detergent was taken as 100%. Results indicated that the lipase was compatible with every detergent that was used in the experiment. However, enzyme activity is increased with the detergent surf excel at a wide range of temperatures ranging from 5 °C to 45 °C, with the highest at 30 °C. At 5 °C, 10 °C, 20 °C, 37 °C and 45 °C, the enzyme retained more than 90% of its activity with Tide and had a maximum enhanced activity of about 11% at 30 °C. The enzyme was compatible at low temperatures with the detergent Ariel, with declines in activity of about 23% and 26% at 30 °C, 37 °C, and 45 °C, respectively.

The lipase shows a positive result when combined with detergent to remove vegetable oil stains. Almost complete removal (very faded stain) of the vegetable oil stain was observed in the cloth dipped in a combination of detergent and the purified lipase enzyme. These findings indicate that *B. safensis* TKW3 lipase is suitable for use in detergent formulations.

## DISCUSSION

Extreme habitats are believed to harbor novel, unexplored microbial species and may act as a source of novel genes capable of withstanding extreme environmental conditions (*Krüger et al., 2018*). Among the numerous extremities, brackish environments have recently captured significant interest due to their role as habitats for diverse halophiles, which represent potential reservoirs of extremozymes. Enzymes from the extremities are increasingly recognized as an important component of their adaptability. The hunt for extremophilic enzymes with potential use in biotechnological sectors is becoming more intense. The aim of this study was to conduct a comprehensive purification and biochemical characterization of the intracellular lipase derived from B. *safensis* TKW3, a moderate halophile isolated from a water sample, collected from the brackish lake Tso Kar.

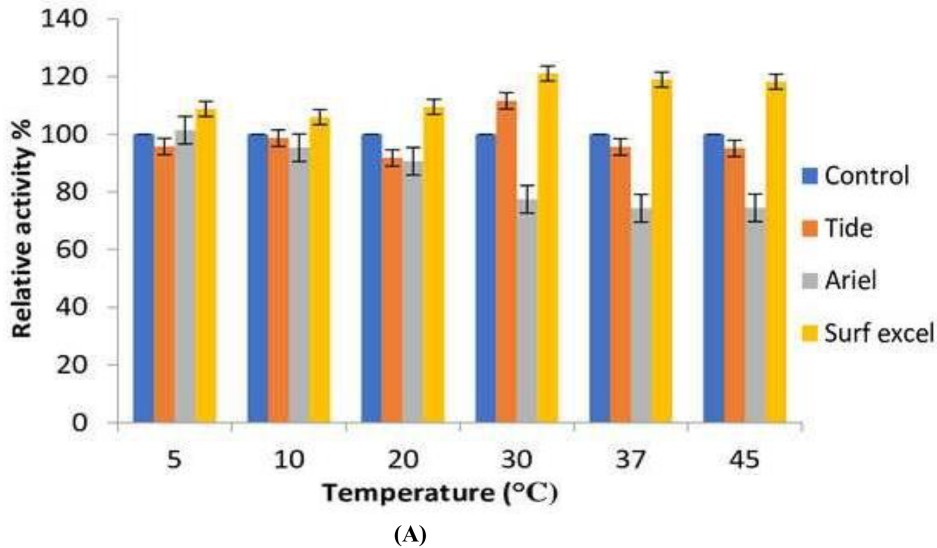

(A)

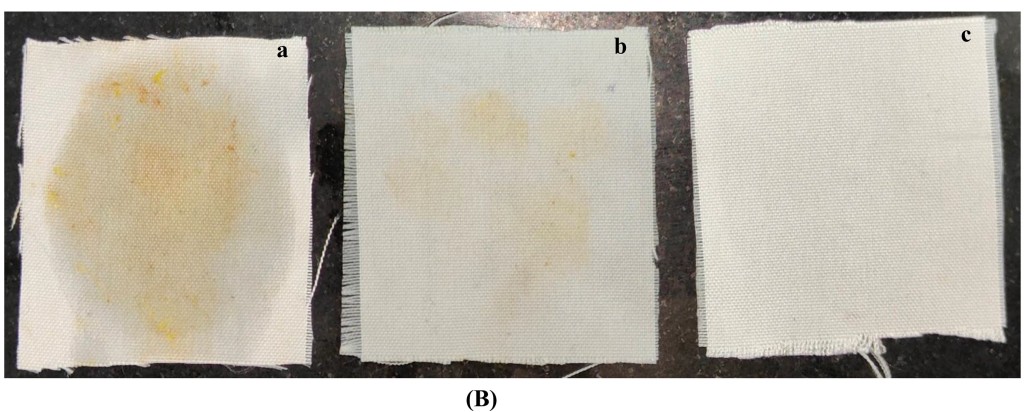

(B)

**Figure 8 Detergent compatibility of *B. safensis* TKW3.** (A) All experiments were performed in triplicates and standard deviation was calculated. The compatibility of *B. safensis* TKW3 lipase with various commercial laundry detergents at different temperatures was checked. Lipase activity without detergent was set as 100%. Surf Excel exhibited the highest activity across a wide temperature range (5 °C to 45 °C), with peak activity at 30 °C. Tide maintained over 90% activity at all temperatures except at 30 °C where it showed an 11% enhancement. Ariel exhibited compatibility at lower temperatures but showed decreased activity at higher temperatures. (B) Washing performance of *B. safensis* TKW3 lipase on oil-stained cloth pieces: (a) cloth washed with distilled water (control), (b) cloth stained with vegetable oil washed with distilled water and detergent, (c) cloth stained with vegetable oil washed with distilled water, purified lipase & detergent.

The purification of the protein involved salting out and dialysis techniques, followed by subsequent purification through hydrophobic interaction chromatography. The lipase from *B. safensis* TKW3 was purified to a degree of roughly 12.01-fold, resulting in a yield of 29.9%. The molecular weight of the purified lipase was determined to be about 28 kDa. Similar to the present study, lipase from *Bacillus* strain NK13 was purified to a single band of 23 kDa (*Zhang et al., 2005*). A 25 kDa lipase was successfully purified from *Halobacillus* sp. AP-MSU with 10.6-fold purity and a 25% yield (*Esakkiraj et al., 2016*). Lipase from B. *safensis* KKSC12 was purified to 55 kDa with a 16.1% (*Patel & Parikh, 2022*).

*B. thermoleovorans* CCR11 was found to produce an enzyme with a molecular weight of about 11 kDa (*Castro-Ochoa et al., 2005*).

The biochemical characterization of the enzyme revealed that the purified lipase exhibited peak activity at 30 °C and maintained stability within the temperature range of 5 °C to 55 °C. The lipase retained 63% activity at 10 °C, and a substantial drop-in activity was observed above 55 °C. Studies have reported that cold-adapted enzymes typically exhibit their peak activity at temperatures below 40 °C and demonstrate heightened sensitivity to heat (*Siddiqui & Cavicchioli, 2006*; *Russell et al., 1998*). The stability of lipase from *Bacillus coagulans* ZJU318 decreased remarkably at 50 °C and 60 °C (*Lianghua & Liming, 2005*). Comparable findings were observed for the cold-adapted lipase derived from *Oceanobacillus* strain PT-11, which exhibited optimal activity at 30 °C and maintained stability up to 50 °C (*Jiewei et al., 2014*). Based on these findings, we inferred that the purified lipase is a cold-adapted lipolytic enzyme. The lipase from *P. lipolyticum* sp. nov., which is adapted to cold conditions, exhibited its peak activity at a temperature of 25 °C. Furthermore, the enzyme demonstrated stability across a broad temperature range of 5–25 °C (*Ryu et al., 2006*). The activity of lipase is significantly affected by pH. In this study, the purified lipase was assessed for its activity across a pH spectrum from 4 to 12. Results indicated an optimal pH of 9.0 for the purified enzyme, with stability observed within the pH range of 7.0 to 10.0. At pH 11.0, the enzyme exhibited a notable decrease in stability, retaining only approximately 12% of its activity. These findings affirm the enzyme's alkaline characteristics. Many industries are currently investigating alkaline lipases for their potential in a variety of commercial applications. Lipase activity of *B. thermoleovorans* CCR11 (*Castro-Ochoa et al., 2005*), *Bacillus cereus* C71 (*Shaoxin, Lili & Bingzhao, 2007*), *Halobacillus* sp. (*Esakkiraj et al., 2016*), and *Bacillus coagulans* ZJU318 (*Lianghua & Liming, 2005*) was reported to be highest at pH 9.0, which is consistent with the results of the current study. Similar to the present study, extracellular lipase from *B. coagulans* ZJU318 was stable over a pH range of 7.0–10.0. Purified lipase is both alkaliphilic and psychrotolerant due to its remarkable stability across a broad pH and temperature range. The purified lipase enzyme from *B. safensis* TKW3 isolated from brackish Lake Tso Kar showed optimum lipase activity at 3 M NaCl concentration and a decrease in activity after 3 M salt concentrations. However, the enzyme was found to be stable at all the concentrations of salts examined. The current findings revealed that the lipase enzyme from moderately halophilic bacteria is halophilic in nature. Supporting this finding, reported lipases by *Halobacillus* sp. (*Esakkiraj et al., 2016*) and *Natronococcus* sp. (*Boutaiba et al., 2006*) had the optimum enzyme activity at 2.5 and 4.0 M NaCl concentrations, respectively. It is well documented that surfactants increase the area of the water-lipid interface by decreasing the interfacial tension between water and oil, thereby increasing the rate of reactions catalyzed by lipase (*Shaoxin, Lili & Bingzhao, 2007*). In the present study, the purified lipase showed enhanced activity with the addition of all detergents tested except Tween-20, for which the enzyme showed a residual activity of 73.5%. Tween-20 has been reported to inhibit the lipase purified from *Acinetobacter haemolyticus* TA 106 (*Jagtap & Chopade, 2015*) and from *B. thermoleovorans* CCR11 (*Castro-Ochoa et al., 2005*). Similar to the present study, lipases from *Staphylococcus* sp.

(*Chauhan, Chauhan & Garlapati, 2013*), *Aspergillus* sp. (*Saisubramanian et al., 2006*), and *Rhizopus* sp. (*Derewenda et al., 1994*) also showed improved stability in the presence of SDS and Tween-80. The results of this study suggest that the isolated lipase displays a propensity for metal activation. While the enzyme's activity is not strictly dependent on metal ions, the presence of specific metals enhances its enzymatic performance. The purified lipase activity was not much affected in the presence of the various metal ions examined except for $Cu^{2+}$, which inhibited the activity of the enzyme. An increase in lipase activity in the presence of $Na^+$, $Li^+$, and $K^+$ has been reported for the moderately halophilic bacterial strain *Oceanobacillus rekensis* PT-11 (*Jiewei et al., 2014*). Similar declines in activity with $Cu^{2+}$ have been reported for lipase from *B. cereus* C71 (*Shaoxin, Lili & Bingzhao, 2007*), *Bacillus* sp. (*Ghori, Iqbal & Hameed, 2011*), and *B. coagulans* ZJU318 (*Lianghua & Liming, 2005*). Stimulatory effects of $Mg^{2+}$ and $K^+$ have been reported from *Bacillus* sp. (*Ghori, Iqbal & Hameed, 2011*), and *B. coagulans* BTS-3 (*Kumar et al., 2005*). In this study, it was observed that β-mercaptoethanol and dithiothreitol exerted inhibitory effects on lipase activity, while the enzyme demonstrated tolerance to guanidine hydrochloride. Complete loss of enzyme activity in the presence of β-mercaptoethanol, indicating the presence of the thiol group that is crucial for the catalytic function. The observed decline in activity implies the presence of disulfide bonds within the *B. safensis* TKW3 lipase, which are susceptible to reduction by β-ME and DTT. The investigation highlights the pivotal role of disulfide bridges in enabling the activity of the *B. safensis* TKW3 lipase. However, contrasting findings exist, such as the enhanced activity of the lipase from *Bacillus smithii* BTMS 11 in the presence of β-mercaptoethanol, as reported by *Lailaja & Chandrasekaran (2013)*. Furthermore, the enzyme's activity was boosted by EDTA, a metal chelator. This aligns with observations on the *estSL3* gene from *Alkalibacterium* sp. SL3, where resistance to various chemicals, including β-mercaptoethanol and EDTA, was noted, as reported by *Wang et al. (2016)*. Similar effects of EDTA have been reported on lipase from *Bacillus* sp. by *Handelsman & Shoham (1994)*. Moreover, the *B. safensis* TKW3 lipase displayed optimal activity with medium to long chain fatty acyl esters, particularly p-nitrophenyl octanoate (C8), indicating its classification as a lipase rather than an esterase. This corroborates findings for lipase from *Bacillus. subtilis* 168 (*Lesuisse, schanck & colson, 1993*). Considering the detergent industry's reliance on microbial hydrolytic enzymes, our study examined the compatibility of the lipase with detergents. We found that the lipase remained stable over a wide temperature range (5 °C to 45 °C), making it advantageous for low-temperature washing, which conserves energy. This stability aligns with reports of enhanced lipase activity in *Staphylococcus arlettae* JPBW-1 with a commercial detergent wheel by *Chauhan, Chauhan & Garlapati (2013)*, and the stability of detergent lipases from *T. asahii* MSR 54 and Cryptococcus sp. S-2 reported by *Suresh Kumar et al. (2009)* and *Thirunavukarasu et al. (2008)*, respectively. Our comprehensive examination of the moderately halophilic *B. safensis* TKW3 lipase confirms its suitability for detergent applications across diverse temperature ranges, showcasing remarkable stability even under challenging conditions. This suggests its potential as an additive in commercial detergents, substantiating prior research while presenting enhanced efficiency in our study. In conclusion, the findings

underscore the significant potential of the *B. safensis* TKW3 lipase in revolutionizing detergent formulations.

## CONCLUSIONS

In this study, we isolated an intracellular enzyme from moderately halophilic bacteria found in the brackish lake Tso Kar in Ladakh. The intracellular lipase from *B. safensis* TKW3 was successfully purified and subjected to thorough biochemical characterization, unveiling a multitude of properties that render it highly suitable for various industrial applications. Notably, the enzyme displayed a cold-adapted nature, with optimal activity observed at 30 °C, and exhibited alkaline characteristics, necessitating 3 M NaCl for activity. These findings strongly suggest that this enzyme is a low temperature-adapted, halophilic, alkaline lipase. With its demonstrated stability across a wide range of temperatures and pH levels, as well as its resilience to NaCl, metal ions, and surfactants, this lipase enzyme emerges as a promising candidate for diverse industrial applications, notably in detergent manufacturing. Moreover, the intracellular lipase from *B. safensis* TKW3 showcased enhanced stability in surfactants and commercial detergents, positioning it as a viable option for inclusion in laundry detergent formulations spanning various temperature conditions.

### Funding
This work was funded by the Council of Scientific and Industrial Research (CSIR), New Delhi, Government of India under grant number 37 (1545)/12/EMR-II entitled "Exploring microbial diversity and Mining Novel Hydrolases from Brackish Water Lakes of the Ladakh Region by Metagenomic Approach". The funders had no role in study design, data collection and analysis, decision to publish, or preparation of the manuscript.

### Grant Disclosures
The following grant information was disclosed by the authors:
Council of Scientific and Industrial Research (CSIR), New Delhi, Government of India: 37 (1545)/12/EMR-II.

### Competing Interests
Srinivas Sistla is an Academic Editor for PeerJ.

### Author Contributions

- Tishu Devi conceived and designed the experiments, performed the experiments, analyzed the data, prepared figures and/or tables, authored or reviewed drafts of the article, and approved the final draft.
- Srinivas Sistla conceived and designed the experiments, analyzed the data, authored or reviewed drafts of the article, and approved the final draft.
- Rabiya T. Khan analyzed the data, prepared figures and/or tables, authored or reviewed drafts of the article, and approved the final draft.
- Swadha Kailoo performed the experiments, prepared figures and/or tables, authored or reviewed drafts of the article, and approved the final draft.
- Mansavi Bhardwaj performed the experiments, prepared figures and/or tables, authored or reviewed drafts of the article, and approved the final draft.
- Shafaq Rasool conceived and designed the experiments, analyzed the data, authored or reviewed drafts of the article, and approved the final draft.

### Data Availability

The raw data are available in the Supplemental Files.

### Supplemental Information

Supplemental information for this article can be found online at http://dx.doi.org/10.7717/peerj.18921#supplemental-information.

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
