# Peer review of "Purification and characterization of detergent stable alkaline lipase from Bacillus safensis TKW3 isolated from Tso Kar brackish water lake"

_PeerJ, doi:10.7717/peerj.18921_

## Round 0.1 · original submission · Major Revisions

The reviewers have now commented on this article and they have raised few major concerns on this manuscript. Therefore, there is need to make major revisions to address those concerns.

Reviewer 1 ·

Basic reporting

English of this manuscript is poor. Sometimes, it might become difficult for the reader to understand what they wanted to say.
Relevant previous literature is appropriately cited, but more recent literature sources could have been used.

Experimental design

In many parts of the material methods, metods essential for the reproducibility of the experiments is missing.
Examples include the following,
-substrate concentrations, enzyme concentrations, temperature and pH values of the conditions for enzyme assay,
-duration of the enzyme activity assay,
-preparation of the standard pNP standard graph used in enzyme activity calculation,
- temperature at which dialysis was performed

Substrate specificity test:
- concentrations for substrates are not specified
- No Km value (or Vmax) was determined for the substrate in the standard assay or for any other substrate.

Validity of the findings

Line 217: It is stated that there is an increase in enzyme production, but it is not clear which medium/condition is compared with the medium in this study.
One of the most important shortcomings of the paper is that the purified and characterized lipase enzyme was not used in any washing experiment to remove oils for validity.

Additional comments

I have indicated below the parts that need to be corrected.

Introduction:
-In sentences containing more than two examples, a comma should be placed before “and”. E.g. Halophiles are regarded as a treasure trove of new biomolecules, metabolites and biomaterials.

-Words that should not be capitalized are capitalized. E.g. Halozyme, Lipase

Material Method
-Reference should be made to the previous study from which the strain was isolated.

-There is no mention of the bacteria's extracellular enzyme.

-The accession number of the bacterial strain should be updated as OP735582.1

- There must be a space between the unit and number used e.g. 3 M, 30 °C

- Line 136: kDa

-The use of only a single time period (30 minutes) in enzyme stability tests is a shortcoming.

-It is interesting to observe no activity at pH 6. Because pH 6 is not a very acidic environment

-There are problems with the spelling of the genus names of bacteria. Those that should be abbreviated are written in expanded form, and those that should be written in expanded form are written in abbreviated form. E.g. Bacillus smithii, P. lipolyticum

Reviewer 2 ·

Basic reporting

In this manuscript, the authors purified the lipase enzyme with a molecular mass of 28 kDa from B. safensis TKW3, exhibiting stability across a wide temperature range (5°C to 55°C) and a broad pH range (7.0 to 9.0). The purified lipase showed promise for applications in the leather and detergent industries, demonstrating compatibility with various pH values, surfactants, metal ions, and inhibitors. Overall, the data presented are interesting and valuable to the field. While the manuscript was not written well and some of the data presented are pretty simple, need to do a better interpretation and discussion. The authors are advised to meticulously review and revise the manuscript.

Major or reject

1. No page number
2. The authors need to avoid long sentences or split long sentences into several short sentences. Like: L17-22
3. L44 need not stop punctuation before a reference.
4. Overall, no gene and/or protein sequence information about this lipase. If the authors can do some data analysis about gene or protein search would be great for this manuscript.
5. L215-220 Production of lipase, need to add more data and show them with column figures.
6. L222-230: The authors need to run all eluents from HiLoad Superdex column. I was really concerned about the purity of lipase. This part is the key point of this manuscript, the authors need to add more data and discuss more here.
7. Fig. 2-4 can be merged as one figure. Fig. 5 and 6 can also be merged as one.
8. Fig. 2-8: The authors need to add figure legends for Fig. 2-8. The current simple description is not enough for readers to understand your comparison.
9. For several figures, such as Fig. 5-8, statistical analysis is needed.

Experimental design

as listed in 1

Validity of the findings

as listed in 1

Additional comments

as listed in 1

Reviewer 3 ·

Basic reporting

This study is quite well designed, and the manuscript is well written and easy to read.

Experimental design

1. The identification of the potent isolate should be elaborated, including primers used for PCR and PCR conditions. Moreover, the authors should mention the analysis of nucleotide sequences and phylogenetic tree, including the software and the method.

Validity of the findings

The results are well described, but phylogenetic tree should be added.

---

## Round 0.2 · Minor Revisions

There is need to address the questions raised by the reviewers.

Reviewer 1 ·

Basic reporting

Literature references are not visible.
Figures should be corrected.

Experimental design

Necessary corrections have been made

Validity of the findings

Necessary corrections have been made.

Additional comments

There are still grammatical errors that I pointed out in my previous review.
-Space between number and unit e.g. 3 M, 30 °C,
-spelling of the bacteria names should be corrected (e.g. B.cereus, B.safensis, B. safensis are wrong),
-use mL or ml throughout the manuscript
-KDa should be kDa.


In Figure 1. There is no unit for pNP concentration in the standard curve provided
In Figure 5, Figure 6 and Figure 7. The titles (and units) for y-axis are missing.
In Figure 7. The unit for salt concentration is missing
In Figure 8 and 10. The title for x-axis is missing
There is no Figure 8.b so you should write Figure 8 instead of Figure 8.a.

In Figure 11: The control (cloth washed with water) was compared to washing with a solution containing enzyme + detergent. The observed cleaning effect may be due to the detergent. Results should be given in such a way that the readers can evaluate the effect of enzyme (You can make use of the representation of the results in the reference study DOI: 10.1016/j.pep.2021.105819).

When specifying figures in the text, you should write Figure 11.a.

Reviewer 2 ·

Basic reporting

The authors have responded to my comments well. Only two minor comments need to be addressed: 1) Too many figures in the main text, the authors need to merge or move several of them to supplementary. 2) Statistical analysis method should be noticed in each figure legend.

Experimental design

as listed in 1

Validity of the findings

as listed in 1

---

## Round 0.3 · accepted · Accept

As the authors have addressed the comments raised by the reviewers, I think this article is acceptable now.